# Factors affecting the relapse of maxilla and soft tissues of nose, upper lip and velopharyngeal structures after maxillary advancement in cleft patients

**Sirada Chaisiri[1], Raweewan Arayasantiparb[2], Kiatanant Boonsiriseth[1]***

**1** Department of Oral and Maxillofacial Surgery, Faculty of Dentistry, Mahidol University, Nakhon Pathom, Thailand, **2** Department of Oral and Maxillofacial Radiology, Faculty of Dentistry, Mahidol University, Nakhon Pathom, Thailand

\* kiatanant.boo@mahidol.ac.th

**Data Availability Statement:** All relevant data are within the manuscript.

## Abstract

The objectives of this study were to find the factors affecting the relapses after maxillary advancement in cleft patients. This retrospective study included 25 cleft patients. The serial lateral cephalograms were used for the evaluation of the maxilla and the soft tissue relapses in 1-year post-operative period. The skeletal relapse rate correlated with the amount of skeletal movement only in horizontal direction (r = 0.483, p = 0.015). The patients with significant skeletal relapse did not have different amount of soft tissue relapses when compared to the patients without significant skeletal relapse, except for the A' point. Relapses of the soft tissue parameters of the upper lip and nose were correlated with the upper incisor tip position horizontally and vertically. When comparing the patients who underwent maxillary distraction (DO) with the patients who underwent conventional orthognathic surgery (CO), the DO group had greater NLA relapse and increase of pharyngeal depth than the CO group.

## Introduction

Cleft lip and palate (CLP) patients have an intrinsic deficiency in the midfacial skeleton that is made worse by operations [1]. To achieve adequate aesthetic and functional results, 11.2–54.7% of CLP patients require orthognathic surgery [1–4]. One of the most common problems after orthognathic surgery in CLP patients is post-operative relapse. Relapse rates of 9–31% horizontally, 18.8–52% vertically, and 4.5–30% rotationally were reported [5–11].

Several factors were evaluated for their possible influence on relapse. The cleft type was occasionally reported to be associated with relapse. Bilateral CLP patients were reported to have more skeletal relapse than unilateral CLP patients [8]. A recent systematic review conversely reported that unilateral cases had a higher relapse rate [12]. Other studies did not find any relationships [9, 13]. It is inconclusive whether the cleft type is associated with the skeletal relapse. The amount of surgical movement was extensively studied for its relationship with the skeletal relapse. Some studies found positive correlations between them in all directions, including horizontal, vertical, and rotational [5, 8]. On the other hand, a study found no

**Funding:** The author(s) received no specific funding for this work.

**Competing interests:** The authors have declared that no competing interests exist.

correlation between movement and relapse in both horizontal and vertical directions [14]. A more severe degree of maxillary hypoplasia was common in patients undergoing bimaxillary surgery. However, they had comparable skeletal relapse with those who underwent isolated maxillary surgery [9, 13, 15]. The results may be due to the degree of the mean maxillary advancement in both groups being similar [9]. Occasionally, the bimaxillary group was reported to have a higher skeletal relapse [12]. Distraction osteogenesis (DO) procedures offered a gradual skeletal advancement, allowing larger maxillary advancement with more stability than conventional orthognathic surgery (CO) [16–18]. The skeletal relapse was lower in patients who had Le Fort I (LFI) surgery with interpositional bone grafting [9, 17]. However, a study with mean maxillary advancement of 5.5 mm reported good stability even without bone grafting [10]. Gender, age at surgery, presence of pharyngeal flap, and presence of alveolar cleft were not associated with skeletal relapse [5, 8, 12–14].

Soft tissues of nose, upper lip, and velopharyngeal structures are inevitably affected by maxillary advancement. Soft tissue scarring affected tonicity, thickness, and response of the overlying soft tissue to the movement of the underlying dental and osseous components [19]. The ratios for horizontal advancement of point Pn:A ranged from 0.15:1 to 0.89:1, and the ratios for horizontal advancement of A':A ranged from 0.5:1 to 1.1:1 [10, 20–25]. The correlations between these changes and the amount of skeletal movement are unclear, and other factors may be causing these ratio variations. Different cleft types may affect soft tissue changes and were suggested to be investigated [22]. Movements of the upper incisor tip (UIT) were the most significant determinant of post-operative upper lip position [25]. Differences in facial soft tissue changes between the DO group and the CO group were observed. The DO group had a more consistent and a relatively higher anterior movement ratio of soft to hard tissue [22, 24, 26]. Post-operative increase of pharyngeal depth was expected. The amount of maxillary advancement and method of maxillary advancement (CO or DO) were occasionally mentioned to have effects on velopharyngeal changes. However, inconsistent velar length, and velar thickness changes were found [27–30]. So far, the results of previous studies showed that post-operative changes of soft tissue components in maxillary advancement in cleft patients are scattered. Only a few studies reported possible influencing factors.

This study aims to find the factors associated with skeletal relapse after maxillary advancement in cleft patients. Secondly, we evaluate whether the soft tissue of the nose, upper lip, and velopharyngeal structures relapse correlatedly to the skeletal relapses. Lastly, we aim to find other factors related to different soft tissue relapses post-operatively. Variables including cleft type, presence of cleft palate, amount of surgical movement, isolated maxillary or bimaxillary surgery, CO or DO, and the use of interpositional bone graft are evaluated for their effects on skeletal relapse. For post-operative soft tissue relapse, variables including skeletal relapse, pre-operative facial and velopharyngeal soft tissues, nasal cinch, and position of UIT are also evaluated.

## Materials and methods

A retrospective study was performed. The study had the approval of the Faculty of Dentistry/ Faculty of Pharmacy, Mahidol University, Institutional Review Board COA No. MU-DT/ PY-IRB 2021/041.2104. The study, including data collection, was carried out from May 2021 to September 2022.

### Subjects

This study includes cleft patients who underwent maxillary advancement surgery at the Faculty of Dentistry, Mahidol University from 2010 to 2020, who had been followed up for more

than 1-year post-operatively. Cleft patients included in this study are as follow, unilateral or bilateral cleft lip and/or cleft palate patients, patients with or without other deformities/syndrome associations. Patients who underwent maxillary advancement methods by either conventional orthognathic surgery or by distraction osteogenesis, either with or without simultaneous mandibular surgery were included. The patients who had incomplete data and/ or incomplete radiographs were excluded from this study.

**Sample size determination.** The sample size were determined based on the results of the comparable previous study [8]. The study found a significant correlation between vertical surgical displacement and relapse after 1 year ($r^2 = 0.309$). The sample size can be calculated as follows:

Formula:

$$C(r) = \frac{1}{2} ln[\frac{1+r}{1-r}] = 0.629$$

$$n = [\frac{Z_\alpha + Z_\beta}{C(r)}]^2 + 3$$

Notation:

n = Sample size

r = expected correlation coefficient is $\sqrt{0.309} = 0.556$

$Z_\alpha$ = The probability of type I error (significance level), where α = 0.05 will be equal 1.96

$Z_\beta$ = The probability of type II error (1 –power of the test), where $\beta$ = 0.20 will be equal 0.842

C(r) = Fisher's arctanh transformation

Hence:

$$n = [\frac{(1.96 + 0.842)}{0.629}]^2 + 3 = 22.97 \sim 23$$

Therefore, the required sample size is 23 (n = 23).

## Surgical technique

All patients had pre-surgical and post-surgical orthodontic treatment. The decision whether to perform CO or DO was based on the severity of skeletal deformity analyzed from lateral cephalogram and the restriction of soft tissue scar. For patients underwent CO, standard LFI osteotomy was performed. A prefabricated occlusal splint(s) was used intra-operatively to facilitate placement of the jaw(s). The final splint was wired to maxillary arch wire. Four-point fixation was achieved by miniplates over piriform and zygomaticomaxillary buttresses. Interpositional bone grafts, taken from either the proximal segment of mandible after bilateral sagittal split ramus osteotomy (BSSRO) or from the maxillary osteotomy sites, may be placed at osseous gap at anterior maxilla if needed. In maxillary distraction group, after LFI osteotomy, the external or internal distractors were oriented as planned vectors and fixation was made. Latency period and rate of distraction were adjusted for individual patients. Simultaneous BSSRO was performed only if the mandible required surgical correction. Maxillomandibular fixation also applied for 2 to 6 weeks. Light orthodontic traction might be used to control the occlusion in the early post-operative period.

## Cephalometric analysis

The lateral cephalograms were taken by radiographic machine CS9000C (Carestream Health Inc., Rochester, NY) and recorded in digital imaging and communications in medicine (DICOM). Lateral cephalograms taken at pre-operative (T0), immediate post-operative (T1), 3-month post-operative (T2), and 1-year post-operative (T3) were used. The radiographs taken at the end of distraction were used as immediate post-operative (T1) in patients who underwent maxillary distraction. The one-year post-operative radiographs were used due to the skeletal stability after maxillary advancement was expected [31]. For the soft tissue changes analysis, we chose a 3-month post-operative radiograph (T2) instead of the immediate post-operative radiograph (T1) to avoid errors from immediate post-operative soft tissue swelling. All lateral cephalograms were traced and superimposed using Dolphin® V.11.0 software (Dolphin Imaging and Management Solutions, Chatsworth, Calif.). The horizontal reference line (X-plane) was taken at 7 degrees below the S-N line. A line perpendicular to the X-plane, drawn through the S point was used as a vertical reference line (Y-plane). Horizontal and vertical movements of the maxilla were assessed by measuring the change of point A on the X-plane and Y-plane subsequently. The amount of maxillary rotation was measured as the change of the palatal plane. Clockwise rotation was designated to be positive and counterclockwise rotation was designated to be negative. The landmarks, planes, and angles in this study were modified from comparable previous studies [8, 9, 11, 27, 29]. The following parameters were used for analysis (Table 1, Fig 1).

Serial cephalograms of each patient were superimposed on the anterior cranial reference structures. Surgical movements were determined by measuring differences between T1 and T0. Relapses of skeletal component were determined by measuring differences between T3 and T1. The soft tissue relapses were determined by measuring differences between T3 and T2. Final movements of both hard and soft tissue were determined by measuring differences between T3 and T0. All tracings were repeated twice by one examiner with an interval of more than two months and averages were used for analysis.

## Data analysis

These factors were identified and evaluated for their relationship with the relapse parameters.

Factors associated with skeletal relapse:

The amount of surgical skeletal movement, cleft type, isolated maxillary or bimaxillary surgery, CO or DO, and use of interpositional bone graft were analyzed as factors affecting the

**Table 1. Cephalometric landmarks and reference planes.**

| Points | | Points | |
|---|---|---|---|
| S | Sella turcica | PNS | Posterior nasal spine |
| N | Nasion | Uv | Point of uvular tip |
| A | Subspinale | UPW | Upper pharyngeal wall: junction between posterior |
| Pn | Point of nose | | pharyngeal wall and palatal plane |
| A' | Soft tissue A point | PD | Pharyngeal depth: distance from PNS to UPW |
| Sn | Subnasale | **Planes** | |
| Ls | Labialis superioris | X-axis | S-N line + 7 degrees |
| Cm | Columella | Y-axis | Line drawn through point S, perpendicular to X-axis |
| Stms | Stomium superior | PP | Palatal plane—ANS-PNS |
| UIT | Upper incisor tip | **Angle** | |
| ANS | Anterior nasal spine | NLA | Nasolabial angle |

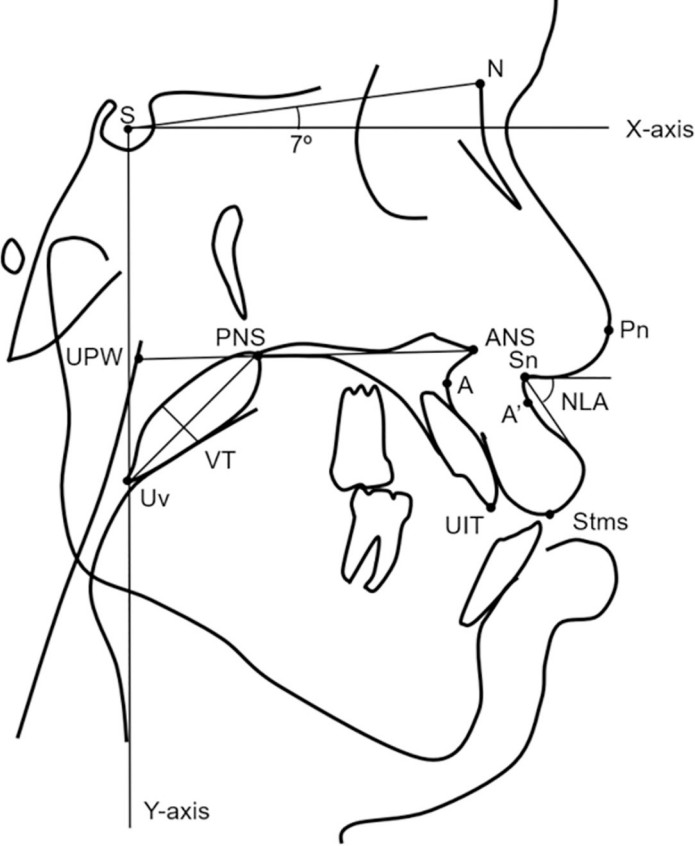

**Fig 1. Tracing of lateral cephalograms.**

skeletal relapse rate. Correlations between the skeletal relapse rate and the amount of skeletal surgical movement were evaluated in all directions. Other factors were evaluated by dividing the patients into two groups, and the mean horizontal skeletal relapse rates were then compared. The relapse rate was the percentage of amount of relapse (mm.) by amount of movement (mm.). The mean relapse rate = (relapse rate$_1$ + relapse rate$_2$ + . . . + relapse rate$_n$)/n [17].

Association between skeletal relapse and soft tissue relapse:

To evaluate whether significant skeletal relapse could affect the associated soft tissues relapses, we compared the differences of the soft tissues relapse between patients who had significant skeletal relapse and who did not. Patients who had significant skeletal relapse had ≥2 mm. of horizontal relapse. The criteria is from a study by Proffit et al [32]. where changes of less than 2 mm were considered within the range of method error and clinically insignificant. Changes of ≥2 mm were outside the range of method error and potentially clinically significant.

Evaluation of soft tissue relapses was divided into 2 parts, the nose and upper lip region, and velopharyngeal region. Changes of Pn, Sn, A', Ls, Stms position, and NLA were measured for nose and upper lip soft tissue evaluation. Changes of velar thickness, velar length, and pharyngeal depth were measured for velopharyngeal soft tissue evaluation. The association between relapses of skeletal and soft tissues was evaluated only in the horizontal plane.

Factors associated with soft tissues relapse:

Cleft types, CO or DO, nasal cinch suture, and change of upper incisor tip were analyzed as factors affecting post-operative nose and upper lip relapse. CO or DO were analyzed as factors affecting post-operative velopharyngeal relapse. Only correlations between the amount of nose and upper lip soft tissue and the amount of UIT changes were evaluated in both horizontal and vertical directions. Other factors were evaluated by dividing the patients into two groups, and the mean amount of horizontal soft tissue relapses were then compared.

## Statistical analysis

The reliability of the measurements was evaluated by an intra-class correlation coefficient (ICC). For factors investigation, correlations between two continuous variables were examined by Pearson's correlation or Spearman's rho correlation. The differences between groups were compared using the t-test or Mann-Whitney U test. The ANCOVA analysis was used to adjust the influences of potential confounding variable on the dependent variables. The level of significance was set at $P<0.05$. IBM software package SPSS Statistics V.15 (SPSS Inc, Chicago, IL, USA) was used for statistical analysis.

## Result

Twenty-five cleft patients who underwent maxillary advancement were included in this study. Two patients had isolated cleft palate and 23 patients had cleft lip and palate. Nineteen patients had unilateral CLP and 4 patients had bilateral CLP. All patients had cheiloplasty, palatoplasty, and alveolar bone grafting if indicated before maxillary advancement surgery. The mean age at the maxillary advancement operation was 22.0±4.6 years. Population data were shown in Table 2.

For the evaluation of the reliability of the measurements, the results showed that the operator was consistent in the repeated cephalometric measurements. The ICC values were > 0.9 for all cephalometric parameters, indicative of very high intra-examiner reliability (Table 3).

### Factors affecting the skeletal relapse

Relapse of skeletal component was determined by measuring differences between T3 and T1. Mean horizontal advancement from surgery was 3.8±2.8 mm (ranged from 0 to 9.8 mm.), and

**Table 2. Study population data.**

| Factors evaluated in this study | | n = 25 |
|---|---|---|
| **Cleft type** | Isolated CP | 2 |
| | Unilateral CLP | 19 |
| | Bilateral CLP | 4 |
| **Isolated maxillary or bimaxillary surgery** | Isolated maxillary | 5 |
| | Bimaxillary | 20 |
| **Maxillary advancement method** | Conventional orthognathic surgery | 19 |
| | **Interpositional bone graft** | |
| | No | 5 |
| | Yes | 14 |
| | Distraction osteogenesis | 6 |
| **Nasal cinch** | No | 15 |
| | Yes | 10 |

**Table 3. Intra-class correlation coefficient (ICC) and absolute difference (mean±SD) between the two measurements of the examiner.**

| Landmarks | ICC | Absolute difference | | Landmarks | ICC | Absolute difference | |
|---|---|---|---|---|---|---|---|
| | | Mean | SD | | | Mean | SD |
| A point (horizontal) | 0.999 | 0.39 | 0.03 | Ls point (vertical) | 0.988 | 0.59 | 0.05 |
| A point (vertical) | 0.990 | 0.68 | 0.11 | Stms point (horizontal) | 0.994 | 0.74 | 0.05 |
| Palatal plane | 0.991 | 0.62 | 0.06 | Stms point (vertical) | 0.996 | 0.67 | 0.20 |
| Pn point (horizontal) | 0.997 | 0.39 | 0.03 | NLA | 0.998 | 1.07 | 0.07 |
| Pn point (vertical) | 0.997 | 0.57 | 0.05 | UIT point (horizontal) | 0.999 | 0.33 | 0.03 |
| Sn point (horizontal) | 0.996 | 0.51 | 0.11 | UIT point (vertical) | 0.990 | 0.43 | 0.06 |
| Sn point (vertical) | 0.996 | 0.50 | 0.05 | Velar thickness | 0.949 | 0.73 | 0.05 |
| A' point (horizontal) | 0.996 | 0.41 | 0.04 | Velar length | 0.995 | 0.55 | 0.04 |
| A' point (vertical) | 0.996 | 0.34 | 0.03 | Pharyngeal depth | 0.986 | 0.73 | 0.05 |
| Ls point (horizontal) | 0.996 | 0.41 | 0.04 | | | | |

mean relapse was 1.45±1.69 mm (ranged from -0.5 to 5.1 mm). The maxillary height was shortened up to 5.3 mm (mean 2.4±1.8 mm.) in 6 patients and was elongated up to 13.4 mm (mean 4.4±3.5 mm.) in 19 patients. A mean vertical relapse of 0.6±1.2 mm. was found in vertically impact maxilla. A mean vertical relapse of 2.0±2.0 mm. was found in inferior positioning maxilla. For the change of the palatal planes, 20 patients had clockwise maxillary rotation (mean 5.6±4.3 degrees, ranged from 0.5 to 15.4 degrees) with mean relapse of 2.4±4.9 degrees. Five patients had counterclockwise maxillary rotation (mean 4.8±2.2 degrees, ranged from 1.7 to 7.6 degrees), instead of relapse, a mean of 2.3±3.2 degrees further counterclockwise movement was found in this group.

There is moderate positive correlation between amount of horizontal skeletal movement and amount of horizontal skeletal relapse rate (r = 0.483, p = 0.015). A moderate negative correlation between amount of rotational skeletal movement and amount of rotational skeletal relapse rate was found (r = -0.469, p = 0.018). No correlations were found in vertical direction (Table 4, Figs 2 and 3).

The ANCOVA analysis was used to adjust the influences of the amount of advancement when comparing the horizontal skeletal relapse rates between two groups of patients with different factors. No significant difference was seen between unilateral and bilateral CLP patients, or between patients who underwent isolated maxillary surgery and patients who underwent two-jaw surgery, or between those who underwent CO and those who had maxillary DO, or between those who had interpositional bone graft and those who did not. Additionally, the amounts of skeletal advancement were compared between groups for further discussion (Table 5).

**Table 4. Correlation between amount of skeletal surgical movement and skeletal relapse rate.**

| Correlation between amount of skeletal surgical movement and skeletal relapse rate | | | |
|---|---|---|---|
| | Horizontal | Vertical | Rotational |
| r | **0.483**[*] | 0.061 | **-0.469**[*] |
| P-value | **0.015** | 0.773 | **0.018** |

Data were analyzed with Spearman's rho correlation, r = correlation coefficient

[*]Statistically significant at P<0.05

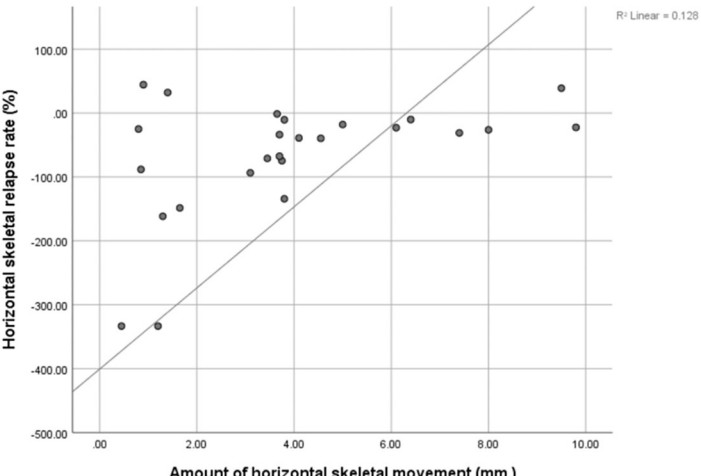

**Fig 2. Significant correlation between the amount of horizontal skeletal movement and the horizontal skeletal relapse rate.**

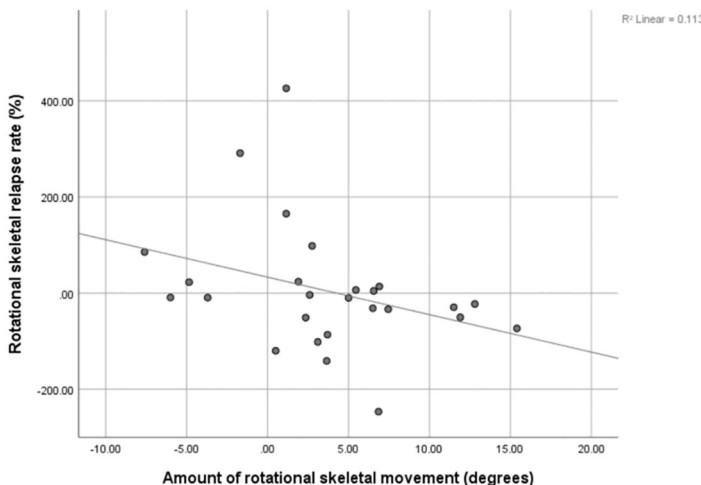

**Fig 3. Significant correlation between the amount of rotational skeletal movement and the rotational skeletal relapse rate.**

## Association between the skeletal relapse and soft tissue relapses

The changes between T2 and T3 of both the skeletal and soft tissues were analyzed. Six patients (24%) had significant skeletal relapses. The patients who had significant skeletal relapse had a relapse of the A' point more than the patient without significant skeletal relapse (P = 0.013). A significantly larger increase in pharyngeal depth was found in patients who had a significant skeletal relapse (P = 0.048) (Table 6).

## Factors affecting the soft tissue relapse

The changes between T2 and T3 were analyzed. Relapses of the nose and upper lip soft tissue after maxillary advancement were not different between unilateral (n = 19) and bilateral

**Table 5. Compare horizontal skeletal relapse rates between two groups of patients with different factors.**

| Factors affecting skeletal relapse rate | Groups | Case | Horizontal skeletal relapse rate (percent) | |
|---|---|---|---|---|
| | | | Median (IQR) | P-value |
| **Cleft types** | Unilateral | 19 | -39.02 (-148.48, -18.00) | 0.902 |
| | Bilateral | 4 | -45.01 (-72.90, 27.72) | |
| **Isolated maxillary or bimaxilllary surgery** | Isolated maxillary | 5 | -22.95 (-28.67, 8.25) | 0.274 |
| | Bimaxillary | 20 | -67.29 (-144.91, -12.40) | |
| **Maxillary advancement method** | CO | 19 | -67.57 (-148.48, -10.53) | 0.377 |
| | DO | 6 | -24.60 (-56.86, -7.10) | |
| **Interpositional bone graft (In CO group)** | Yes | 5 | -93.55 (-1330.00, -17.58) | 0.127 |
| | No | 14 | -53.30 (-103.30, -10.44) | |

Data were presented by median (IQR) and analyzed with Analysis of Covariance (ANCOVA)

Covariate variable is amount advance.

Statistically significant at *P<0.05

(n = 4) CLP patients, or between patients who had nasal cinch (n = 10) and who did not (n = 15) (Table 7). The maxillary advancement method (CO or DO) was evaluated in both the relapses of soft tissues of nose and upper lip, and velopharyngeal region. Patients who underwent maxillary DO (n = 6) had a significantly larger relapse of NLA and larger increase of pharyngeal depth when compared to patients who underwent CO (n = 19) (Table 8). The amounts of skeletal advancement were also compared between groups for further discussion.

The nose and upper lip soft tissues relapse seem to correlate with the position of upper incisor tip. Moderate correlations between the amount of horizontal UIT changes and horizontal changes of Pn and Stms points were found. We also evaluated in vertical planes and found

**Table 6. Compare soft tissues relapses in patients with and without significant skeletal relapse.**

| | | Significant skeletal relapse | | P-value |
|---|---|---|---|---|
| | | Yes (n = 6) | No (n = 19) | |
| | | Mean±SD/Median (IQR) | Mean±SD/Median (IQR) | |
| Amount of skeletal advancement from surgery (mm.) | | 3.73 (1.28, 7.55) | 3.65 (0.90, 5.00) | ‡0.588 |
| Amount of horizontal skeletal relapse (mm.) | | -2.72±0.74 | -0.33±0.86 | <0.001*** |
| **Nose and upper lip** | ΔPn (horizontal) (mm.) | -0.65 (-1.70, 1.60) | -0.05 (-0.55, 0.80) | ‡0.702 |
| | ΔSn (horizontal) (mm.) | -1.02±2.32 | -0.80±2.60 | 0.855 |
| | **ΔA' (horizontal) (mm.)** | **-2.53(-3.03, 1.40)** | **-0.40 (-0.85, 0.40)** | **‡0.013*** |
| | ΔLs (horizontal) (mm.) | -0.58±2.70 | -0.92±1.64 | 0.970 |
| | ΔStms (horizontal) (mm.) | -1.67±1.79 | 0.38±1.70 | 0.557 |
| | ΔNLA (degree) | -3.85 (-10.78, -1.09) | -4.50 (-7.85, -1.30) | ‡0.975 |
| **Velopharyngeal structure** | ΔVelar thickness (mm.) | 1.48 (-2.10, 0.63) | -0.40 (-1.13, 0.30) | ‡0.205 |
| | ΔVelar length (mm.) | 0.43±5.75 | -0.42±4.05 | 0.692 |
| | **ΔPharyngeal depth (mm.)** | **6.78±1.67** | **2.69±4.62** | **0.048*** |

Data were analyzed with independent T-test, except

‡ using Mann-Whitney U test

Statistically significant at

*P<0.05,

**P<0.01,

***P<0.001

**Table 7. Compare nose and upper lip soft tissues relapses between unilateral and bilateral patients, between the patients who had nasal cinch and who did not.**

| Nose/ upper lip parameters | Cleft type | | | Nasal cinch | | |
|---|---|---|---|---|---|---|
| | Unilateral (n = 19) | Bilateral (n = 4) | P-value | Yes (n = 10) | No (n = 15) | P-value |
| | Mean±SD/ Median (IQR) | Mean±SD/ Median (IQR) | | Mean±SD/ Median (IQR) | Mean±SD/ Median (IQR) | |
| Amount of skeletal advancement (mm.) | | | | | | |
| | 3.45±2.73 | 4.54±3.75 | 0.504 | 4.07±2.69 | 3.54±2.94 | 0.650 |
| ΔPn (mm.) | 0.05 (-0.55, 0.80) | -0.90 (-1.85, 2.68) | ‡0.394 | 0.05 (-0.99, 0.83) | -0.05 (-0.55, 0.80) | ‡0.934 |
| ΔSn (mm.) | -0.83±2.50 | -0.70±3.20 | 0.928 | -1.57±2.11 | -0.37±2.67 | 0.249 |
| ΔA' (mm.) | -0.80 (-1.10, 0.25) | -1.35 (2.53, -0.14) | ‡0.465 | -0.70 (-1.18, 0.65) | -0.55 (-2.40, 0.25) | ‡0.890 |
| ΔLs (mm.) | 0.13±1.86 | -0.46±2.05 | 0.576 | -0.45±1.91 | 0.16±1.89 | 0.445 |
| ΔStms (mm.) | -0.02±1.65 | -0.71±1.88 | 0.463 | -0.41±1.65 | -0.11±1.90 | 0.692 |
| ΔNLA (degree) | -4.50 (-7.85, -2.15) | -6.52 (-14.6, 0.91) | ‡0.745 | -5.58 (-10.93, -1.94) | -3.30 (-5.35, 2.00) | ‡0.279 |

Data were analyzed with using independent T-test, except

‡ using Mann-Whitney U test

* Statistically significant at P<0.05

**Table 8. Compare soft tissue relapses between patients who underwent maxillary advancement by CO or maxillary DO.**

| | | Maxillary advancement method | | P-value |
|---|---|---|---|---|
| | | CO (n = 19) | DO (n = 6) | |
| | | Mean±SD/Median (IQR) | Mean±SD/Median (IQR) | |
| Amount of skeletal advancement (mm.) | | 2.59±1.76 | 7.43±2.24 | <0.001*** |
| **Nose and upper lip** | ΔPn (mm.) | 0.05 (-0.55, 0.90) | -0.50 (-1.90, 0.25) | ‡0.171 |
| | ΔSn (mm.) | -0.36±2.47 | -2.42±1.96 | 0.076 |
| | ΔA' (mm.) | -0.40 (-1.00, (0.55) | -1.10 (-3.03, -0.45) | ‡0.056 |
| | ΔLs (mm.) | 0.28±1.70 | -1.25±2.11 | 0.081 |
| | ΔStms (mm.) | 0.03±1.82 | -1.03±1.46 | 0.208 |
| | **ΔNLA (degree)** | **-2.8 (-5.35, -0.45)** | **-11.4 (-15.78, -4.90)** | **‡0.005**\*\* |
| **Velopharyngeal structure** | ΔVelar thickness (mm.) | -0.53±1.37 | 0.25±1.08 | 0.218 |
| | ΔVelar length (mm.) | 0.03±3.20 | -0.90±7.19 | 0.771 |
| | **ΔPharyngeal depth (mm.)** | **3.10 (1.60, 4.50)** | **8.40 (2.85, 9.93)** | **‡0.031**\* |

Data were analyzed with independent T-test, except

‡ using Mann-Whitney U test

Statistically significant at

*P<0.05,

**P<0.01

moderate correlations between the amount of UIT changes and changes of A' and Ls points, strong correlations between the amount of UIT changes and changes of Sn and Stms points (Table 9).

## Discussion

Many factors were found to be influencing the skeletal relapse after maxillary advancement in CLP patients. However, controversies remained and even with known risk factors, the amount of skeletal relapse was still unpredictable. Even fewer studies reported soft tissue relapses after maxillary advancement in cleft patients. The amount of post-operative relapse and its

**Table 9. Correlation between nose and upper lip soft tissues relapses and changes of upper incisor tip in horizontal and vertical directions.**

| Correlation with ΔUIT during T3-T2 | | ΔPn | ΔSn | ΔA' | ΔLs | ΔStms | ΔNLA |
|---|---|---|---|---|---|---|---|
| Horizontal | r | **0.441** | 0.358 | 0.201 | 0.448 | **0.467** | 0.061 |
| | P | ‡**0.027*** | 0.079 | ‡0.336 | 0.025* | **0.019*** | ‡0.773 |
| Vertical | r | 0.386 | **0.633** | **0.409** | **0.574** | **0.706** | -0.130 |
| | P | 0.057 | ‡**0.001**** | ‡**0.043*** | ‡**0.003**** | <**0.001***** | ‡0.536 |

Data were analyzed with Pearson correlation, except

‡using Spearman's rho correlation.

r = correlation coefficient, p = P-value

* Statistically significant at *P<0.05, **P<0.01, ***P<0.001

associated factor(s) are still varied among studies. A small number, but large diversity of patients combined with different evaluation methods are the main reasons. Despite ensuring reliability in the method of evaluation, the limitation of the number of populations is the main problem in this study. Multi-center study with collaborated evaluation method is recommended to include an adequate number of populations for multivariate analysis.

## Factors affecting skeletal relapse after cleft orthognathic surgery

The amount of advancement was frequently shown to be correlated with the amount of horizontal skeletal relapse in many studies [7, 9, 10]. In our series, the increasing horizontal relapse rates were expected with the increasing amount of advancement. A mean horizontal relapse was 33.8% in this study, which is slightly higher than other relapse rates reported (9–31%) [5, 7, 9–11, 17]. Some studies found correlations in the vertical plane when the vertical skeletal movement was evaluated as maxillary intrusion and inferior repositioning maxilla separately [6, 8]. We found no correlation between the amount of vertical skeletal movement and the vertical skeletal relapse rate. As expected, the maxillary intrusion had a mean vertical relapse rate of 25%, which is lower than the inferior repositioning maxilla (45.5%). We also found a moderate correlation between the amount of rotational skeletal movement and the skeletal relapse rate in rotational direction (r = -0.469, p = 0.018). However, no correlations in rotational plane were found when evaluated as maxillary clockwise and counterclockwise separately. Patients who had clockwise maxillary rotation had a mean rotational skeletal relapse rate of 42.9%. On the other hand, the patients who underwent counterclockwise maxillary rotation gained further counterclockwise skeletal movement (47.9% of counterclockwise rotational movement) during the relapse period ($T_1$-$T_3$). Superior movement of ANS point in the potential gap after LFI osteotomy occurred and resulted in further counterclockwise of the palatal plane in this study.

Other factors were tested by dividing the patients into two groups and comparing the skeletal relapse rates. No significant differences were found in any other factors. A study found a higher relapse rate in bilateral CLP patients. Characteristics of bilateral CLP patients, which include the premaxilla instability, bilateral alveolar clefts, and multiple missing teeth were described [8]. A recent review study found that patients with unilateral CLP had a higher relapse rate [12]. Our study did not find a difference in skeletal relapse between bilateral and unilateral CLP patients. This was agreed by some studies that reported no association between cleft type and relapse rate [6, 9, 13]. Veau classification might not directly describe the true severity of the cleft [9]. Moreover, variations of specific characteristics were not addressed enough in this classification.

**Table 10. Compare skeletal relapse rates and amount of skeletal movement in vertical and rotational directions between patients who had interpositional bone graft and who did not.**

| Interposit-ional bone graft | Vertical | | | | Rotational | | | |
|---|---|---|---|---|---|---|---|---|
| | Skeletal relapse rate (percent) | | Amount of skeletal movement (mm.) | | Skeletal relapse rate (percent) | | Amount of skeletal movement (°) | |
| | Median (IQR) | P | Mean±SD | P | Median (IQR) | P | Mean±SD | P |
| Yes (n = 5) | -83.72 (-131.46, -14.22) | ‡0.139 | -2.92±3.97 | 0.216 | 6.42 (-134.69, 157.43) | ‡0.711 | 5.06±5.45 | 0.227 |
| No (n = 14) | -25.56 (-60.70, 18.41) | | -0.74±3.01 | | -9.73 (-59.92, 41.56) | | 2.08±4.25 | |

Data were analyzed with independent T-test, except

‡ using Mann-Whitney U test

P = P-value,

* Statistically significant at P<0.05

The significantly different amounts of skeletal advancement between the isolated maxillary surgery group and bimaxillary surgery group, and between the CO group and DO group should be mentioned. Despite larger amount of advancement performed in DO group (mean amount of advancement of 7.43 mm. in DO group and 2.59 mm. in CO group), only 24.60% relapse rate occurred. The gradual advancement with DO allows the LFI segment to move a greater distance, overcome the palatal scar tension, and remain stable in its new location [16–18, 33, 34]. In our study, we found a lower relapse rate (22.95%) in the patients undergoing isolated maxillary surgery despite larger advancement (mean amount of advancement of 8.16 mm. in isolated maxillary surgery group and 2.65 mm. in bimaxillary group). However, all of those patients who underwent isolated maxillary surgery in our study were done by distraction (5 of 6 patients who underwent maxillary distraction were operated on as isolated maxillary surgery). Although this study showed no statistically significant results, the lower relapse rate was more likely to result from the positive benefit of maxillary distraction.

We found no significant difference in the amount of relapse between patients who had interpositional bone grafts and who did not. However, the higher horizontal relapse rates in patients who had interpositional bone graft were not expected. A study reported that the less stability of inferior repositioning maxilla and the potential to relapse of surgical rotational movement were found and overcorrection was suggested [8]. The slightly larger amount of inferior repositioning and clockwise rotational movement in patients who had interpositional bone grafts might have an effect on the lower stability in this study (Table 10). The result should not be interpreted as a negative effect of interpositional bone grafting when needed. Interpositional bone grafting in the osteotomy gap was reported to have a positive effect on relapse [9, 17, 34]. To promote healing and stabilize the skeleton, interpositional bone graft in the osteotomy gap if needed still has benefits in CLP patients undergoing maxillary advancement.

### Association between skeletal relapse and associated soft tissue relapse

Significant skeletal relapse could compromise the expected outcomes functionally and esthetically. To our knowledge, no study had evaluated whether significant skeletal relapse could affect the associated soft tissue relapses. We hypothesized that the patients who had significant skeletal relapse would also have differences of the relapses of associated soft tissues when compared to the patients who did not have a significant skeletal relapse. From the results, the difference of the amount of nasal and upper lip relapse was only observed at the A' point. This might be due to the mean horizontal relapse of 2.72 mm. in patients with a significant skeletal

**Table 11. A comparison ratios of final soft tissue changes at 1-year post-operative with previous studies.**

|  | Pn:UIT | Sn:UIT | A':UIT | Ls:UIT | Stms:UIT |
|---|---|---|---|---|---|
| Previous studies [25] | 0.13–0.38 | 0.36–0.66 | 0.17–0.66 | 0.29–0.9 | - |
| Present study | 0.28 | 0.43 | **1.01** | 0.33 | 0.22 |

relapse group. With a larger amount of skeletal relapse, we hypothesized the amount of nasal and upper lip relapse soft tissues relapse would be significantly different.

For the velopharyngeal region, the changes of pharyngeal depth were different between the patients who had a significant skeletal relapse and who did not. Not only the PD further increased while the skeletal relapsed backward, but the patients who had a significant skeletal relapse also had even larger increased pharyngeal depth than the patients who had more skeletal stability. We assumed that the skeletal relapse did not relate to the changes of the pharyngeal depth. Other factors may affect and will be discussed in the next part.

## Associated soft tissues relapse in cleft orthognathic surgery and related factors

**Nose and upper lip soft tissues.** Differences of post-operative soft tissue profile changes between cleft types were only suggested [22]. We did not find differences of the nose and upper lip soft tissue relapse between unilateral and bilateral cleft patients. The type of cleft lip, as mentioned in the previous discussion, only partly correlated with the severity of the cleft. No studies reported an effect of nasal cinch on the nose and upper lip soft tissue relapse.

For practical application, we evaluated the final soft to hard tissue ratio (ST:HT) during T3-T0. The ratios were 0.51:1 for Pn:A, 0.77:1 for Sn:A, 1.8:1 for A':A, 0.33:1 for Ls:UIT, and 0.22 for Stms:UIT. Most ST:HT ratios of nasal and upper lip soft tissues were within ranges from previous studies [25], except A':UIT (Table 11). Despite no differences of nose and upper lip soft tissues relapses between patients who had nasal cinch and who did not, we found higher A':A and A':UIT ratios in patients who had nasal cinch (Table 12). This was agreed by a study reported that their higher Pn:ANS (0.54:1) and A':A ratio (0.68:1) were possibly due to nasal cinch suture [10].

From the results, the difference of the amount of nasal and upper lip relapse between the CO and DO groups was found only at NLA. NLA relapse was significantly larger in the DO group. We also observed significantly greater NLA changes during surgical movement in the DO group (+21.3 degrees in the DO group, +9.2 degrees in the CO group). (P = 0.014) The greater changes of NLA from surgery might improve the soft tissue profile, but a greater risk of relapse might follow in the DO group. However, these dramatic changes and relapses were observed only in NLA in this study.

UIT movements were found to be correlated with some nose and upper lip soft tissue landmarks. A randomized controlled clinical trial study evaluated soft tissue changes after maxillary advancement between CO and DO in cleft patients. Other than the significant differences

**Table 12. A comparison between ratios of final soft tissue to hard tissue movements at 1-year post-operative between patients who had nasal cinch and who did not.**

|  | A':UIT | A':A | Pn:A | Sn:A | Ls:UIT | Stms:UIT |
|---|---|---|---|---|---|---|
| Nasal cinch |  |  |  |  |  |  |
| Yes | **1.45** | **2.30** | 0.33 | 0.64 | 0.39 | 0.29 |
| No | **0.72** | **1.40** | 0.66 | 0.88 | 0.29 | 0.18 |

**Table 13. Correlation between nose and upper lip soft tissues changes and changes of upper incisor tip in horizontal and vertical directions at 1-year post-operative period.**

| Correlation with ΔUIT during T3-T0 | | ΔPn | ΔSn | ΔA' | ΔLs | ΔStms | ΔNLA |
|---|---|---|---|---|---|---|---|
| Horizontal | r | 0.205 | **0.616** | **0.577** | 0.286 | **0.426** | 0.377 |
| | P | 0.326 | **0.001**** | **‡0.003**** | ‡0.166 | **0.034*** | 0.063 |
| Vertical | r | 0.001 | **0.513** | **0.436** | **0.386** | **0.465** | -0.162 |
| | P | ‡0.996 | **‡0.009**** | **‡0.029*** | **‡0.056*** | **‡0.019*** | ‡0.439 |

Data were analyzed with Pearson correlation, except

‡using Spearman's rho correlation.

r = correlation coefficient, p = P-value

* Statistically significant at *P<0.05, **P<0.01, ***P<0.001

found between CO and DO groups, correlations between UIT and soft tissue landmarks were observed. Particularly, the Stms point had a strong correlation with the advancement of UIT at 6-month, 1-year, and 2-year periods [24]. A study found significant correlations between horizontal changes of all soft tissue points and the UIT. Moreover, correlations between Ls and BUL (Stms in our study) were found in the vertical dimension. The author concluded that UIT changes were the best predictor of soft tissue changes [25]. In this study, upper incisor tip position (UIT) was the only factor that correlated with the relapses of the nose and upper lip soft tissues both horizontally and vertically. Other than the relapse period (T3-T2), overall changes at 1-year post-operative period were further evaluated. Three of 6 parameters of nose and upper lip soft tissues (Sn, A', Stms points) moved correlatedly with UIT position in both horizontal and vertical directions (Table 13). Our results agreed that the UIT position significantly determined the nose and upper lip positions. The final position of UIT must be considered in the surgical plan. Changes of UIT position, whether resulting from relapses or post-surgical orthodontic treatment, could significantly affect the nose and upper lip positions, especially in the vertical plane.

**Velopharyngeal soft tissues.** One year after maxillary advancement, we found increased PD, unchanged VL, and unchanged VT after maxillary advancement. The results were consistent with a study that reported an increase in nasopharyngeal depth, unchanged velar length, and an increase in need ratio (NR) after maxillary advancement [29]. Need ratio was calculated by dividing the pharyngeal depth by the velar length. Frequently, it was mentioned in the velopharyngeal configuration cephalometric study as increasing need ratio (NR) after maxillary advancement may be related to the clinical worsening of VPI [29]. When considered as the need ratio, the mean need ratio from pre-operative radiographs was 0.86, and the mean NR from 1-year post-operative radiographs was 1.00 in this study (Table 14).

In this study, a further increase of pharyngeal depth was found in the relapse period in both patients underwent maxillary advancement by CO and DO. Few studies had mentioned the progressive advancement of the maxilla and the soft tissue profile landmarks after maxillary distraction [11, 24]. However, none reported similar results in velopharyngeal structures. Patients who underwent maxillary advancement by DO had significantly larger pharyngeal depth increased than CO group (Table 8). However, it could be caused by patients who had maxillary advancement in the DO group having more amount of advancement than the CO group (mean advancement of 2.59 mm. in CO group, 7.43 mm. in DO group). Because of the progressive increase of PD, the risk of increased need ratio and VPI worsening after maxillary advancement should be cautioned in both groups.

**Table 14. Velopharyngeal soft tissues outcome after maxillary advancement.**

|  | Pre-operative | 1-year post-operative | Changes | P-value |
|---|---|---|---|---|
| **Velar thickness (mm.)** | 9.42 | 9.06 | -0.36 | 0.211 |
| **Velar length (mm.)** | 26.38 | 26.18 | -0.20 | 0.823 |
| **Pharyngeal depth (mm.)** | 22.05 | 25.10 | **+3.05** | ‡**0.001**** |
| **Need ratio (PD/VL)** | 0.86 | 1.00 | **+0.14** | **0.004**** |

Data were analyzed with pair T-test, except

‡ using Wilcoxon signed-rank test

Statistically significant at

*P<0.05,

**P<0.01

## Conclusion

The larger the amount of horizontal skeletal movement, the higher the skeletal relapse rate was observed. Higher skeletal relapse rates were also found in patients who had inferior repositioning and clockwise rotation of the maxilla. The significant, yet relatively small amount of skeletal relapse in this study did not affect the amount of soft tissue relapses, except for the A' point. For nose and upper lip soft tissue relapse, only UIT position significantly affected the position of the upper lip and nose in both horizontal and vertical directions. Post-operative position of the upper incisor tip must be determined as it was the best predictor of the nose and upper lip location. Increase pharyngeal depth and unchanged velar length resulted in an increased need ratio post-operatively. Patients in DO group had larger maxillary advancement and larger relapse of NLA and increase PD than patients who underwent CO.

## Supporting information

**S1 Data.**
(XLSX)

## Author Contributions

**Conceptualization:** Kiatanant Boonsiriseth.

**Data curation:** Raweewan Arayasantiparb.

**Formal analysis:** Sirada Chaisiri.

**Methodology:** Sirada Chaisiri, Raweewan Arayasantiparb, Kiatanant Boonsiriseth.

**Resources:** Kiatanant Boonsiriseth.

**Supervision:** Raweewan Arayasantiparb, Kiatanant Boonsiriseth.

**Writing – original draft:** Sirada Chaisiri.

**Writing – review & editing:** Sirada Chaisiri, Raweewan Arayasantiparb, Kiatanant Boonsiriseth.

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
