## [Decision Letter · Decision Letter 0]

27 Jun 2023

PONE-D-23-08498

Factors affecting the relapse of maxilla and soft tissues of nose, upper lip and velopharyngeal structures after maxillary advancement in cleft patients

PLOS ONE

Dear Dr. Boonsiriseth,

Thank you for submitting your manuscript to PLOS ONE. After careful consideration, we feel that it has merit but does not fully meet PLOS ONE’s publication criteria as it currently stands. Therefore, we invite you to submit a revised version of the manuscript that addresses the points raised during the review process.

ACADEMIC EDITOR: Dear Authors I hope this letter finds you well. We have carefully evaluated your manuscript titled Factors affecting the relapse of maxilla and soft tissues of nose, upper lip and velopharyngeal structures after maxillary advancement in cleft patients, which you submitted for publication in Plose one. While we appreciate the effort and time you have invested in this work, we have decided that the manuscript requires a major revision before we can consider it for publication.One reviewer has provided constructive feedback and identified several areas where the manuscript requires significant improvement. Therefore, we kindly request that you take these comments into consideration and revise your manuscript accordingly. We believe that these revisions will significantly enhance the quality and impact of your work.

We look forward to receiving your revised manuscript.

Kind regards,

Essam Al-Moraissi

Academic Editor

PLOS ONE

Journal Requirements:

Reviewers' comments:

Reviewer's Responses to Questions

**Comments to the Author**

1. Is the manuscript technically sound, and do the data support the conclusions?

Reviewer #1: Partly

Reviewer #2: Yes

2. Has the statistical analysis been performed appropriately and rigorously? 

Reviewer #1: No

Reviewer #2: Yes

3. Have the authors made all data underlying the findings in their manuscript fully available?

Reviewer #1: No

Reviewer #2: Yes

4. Is the manuscript presented in an intelligible fashion and written in standard English?

Reviewer #1: Yes

Reviewer #2: Yes

5. Review Comments to the Author

Reviewer #1: The primary aim of this study was to identify the factors associated with hard-and soft-tissue relapse following maxillary advancement in cleft patients. While the study's objectives are intriguing, there are several concerns that need to be addressed to enhance the quality of the research. In particular, it is important for the authors to consider potential confounding factors and choose appropriate statistical methods. Here are some specific recommendations:

The authors should provide details about the selection method used for distinguishing between distraction osteogenesis (DO) or conventional orthognathic surgery (CO), as well as isolated or bimaxillary surgery. It is crucial to determine if the authors used a uniform subject group for DO and CO procedures. Additionally, the authors should carefully consider potential confounding factors that may influence the outcomes.

In the Method section, the authors refer to "Factor Analysis" in the title. However, factor analysis is a statistical technique used to reduce a large number of variables into a smaller set of factors, and it may not be appropriate for this study. If the authors performed correlation analysis instead, they should clearly state it.

Furthermore, when describing the correlations between variables, it would be helpful to classify the strength of the correlations as strong, moderate, or weak based on the corresponding r values.

Since the authors are examining multiple factors, it is advisable to employ multivariate analysis to identify the variables that significantly influence the relapse rate.

The use of ICC (Intraclass Correlation Coefficient) alone is insufficient to demonstrate reliability. It is recommended to present additional measures such as absolute differences, standard deviation, and coefficient of variation to provide a comprehensive understanding of the reliability of the study findings.

The authors should explain the methodology used to determine the cut-off value for classifying significant relapse or non-significant relapse.

In Table 5, the number of patients appears to be relatively small. It is important to consider the sample size and potential implications for the statistical power of the study.

Reviewer #2: The manuscript explains several factors affecting skeletal and soft tissue relapse in maxillary advancement surgery in cleft patients.

The data presented has been able to justify the findings. Nevertheless, the sample size is not robust enough for the findings to be universally applicable.

Additionally, the data would have been more accurate if a 3-dimensional modality such as CBCT would have been used for measurement.

6. PLOS authors have the option to publish the peer review history of their article (what does this mean?). If published, this will include your full peer review and any attached files.

Reviewer #1: **Yes: **Chihiro Tanikawa

Reviewer #2: No

---

## [Author Response · Author response to Decision Letter 0]

25 Jul 2023

Please kindly refer to the rebuttal letter attached in the response to reviewers section.

---

## [Decision Letter · Decision Letter 1]

8 Aug 2023

PONE-D-23-08498R1Factors affecting the relapse of maxilla and soft tissues of nose, upper lip and velopharyngeal structures after maxillary advancement in cleft patientsPLOS ONE

Dear Dr. Boonsiriseth,

Thank you for submitting your manuscript to PLOS ONE. After careful consideration, we feel that it has merit but does not fully meet PLOS ONE’s publication criteria as it currently stands. Therefore, we invite you to submit a revised version of the manuscript that addresses the points raised during the review process.

We look forward to receiving your revised manuscript.

Kind regards,

Essam Al-Moraissi

Academic Editor

PLOS ONE

Journal Requirements:

**Additional Editor Comments:**

Dear Authors, while you have addressed the comments of reviewers. Reviewer number 1 has raised some comments.

Reviewers' comments:

Reviewer's Responses to Questions

**Comments to the Author**

1. If the authors have adequately addressed your comments raised in a previous round of review and you feel that this manuscript is now acceptable for publication, you may indicate that here to bypass the “Comments to the Author” section, enter your conflict of interest statement in the “Confidential to Editor” section, and submit your "Accept" recommendation.

Reviewer #1: All comments have been addressed

Reviewer #2: All comments have been addressed

2. Is the manuscript technically sound, and do the data support the conclusions?

Reviewer #1: Partly

Reviewer #2: Yes

3. Has the statistical analysis been performed appropriately and rigorously? 

Reviewer #1: No

Reviewer #2: Yes

4. Have the authors made all data underlying the findings in their manuscript fully available?

Reviewer #1: No

Reviewer #2: Yes

5. Is the manuscript presented in an intelligible fashion and written in standard English?

Reviewer #1: Yes

Reviewer #2: Yes

6. Review Comments to the Author

Reviewer #1: The selection of the samples is paramount for this study since the DO group could have severe status according to the reviewer's comments, which can be a bias for the results. At least, the authors should incorporate another statistical analysis to cancel any potential bias. Furthermore, the rationale behind the chosen number of patients must be thoroughly described in the manuscript, rather than simply displaying the figures in the comments.

Reviewer #2: The study attempted to look for factors associated with skeletal relapse after maxillary advancement in cleft patients along with the correlation with soft tissue relapses. The study seems to be properly planned and executed and would serve as a valuable addition to the literature pertaining to cleft and craniofacial surgery.

7. PLOS authors have the option to publish the peer review history of their article (what does this mean?). If published, this will include your full peer review and any attached files.

Reviewer #1: **Yes: **Chihiro Tanikawa

Reviewer #2: No

---

## [Author Response · Author response to Decision Letter 1]

5 Sep 2023

Reviewer #1: 

1. The selection of the samples is paramount for this study since the DO group could have severe status according to the reviewer's comments, which can be a bias for the results. At least, the authors should incorporate another statistical analysis to cancel any potential bias. 

 We thank you again for giving us the opportunity to strengthen our manuscript with your valuable comments and queries. We tried our best to reduce the potential bias from the potential confounding factor “the amount of skeletal advancement”. The ANCOVA analysis was used instead of t-test/Mann Whitney U test to control/reduce the effects of the amount of advancement when comparing skeletal relapse. Not only between CO and DO group, but also other factors were revised. Unfortunately, after ANCOVA analysis was used, no significant difference of skeletal relapse rates between two groups of patients with different factors, including between patients underwent maxillary advancement by CO and DO. 

According to the revision above, we updated table 5 in our manuscript along with minor adjustments in statistical analysis, result, and discussion parts. Please refer to the revision details mentioned below.

1. (Page 9) Statistical analysis part was revised by adding the ANCOVA analysis along with minor context refinements. 

2. (Page 12) In the result part, minor refinement at the introduction sentence before comparing skeletal relapse between groups with ANCOVA analysis. 

3. (Page 13) Table 5 was updated as listed below 

- As the amount of advancement has been included in the ANCOVA analysis and for the simplicity of the table. the amount of skeletal advancement column was removed. The title of the table has been changed from “Compare horizontal skeletal relapse rates and amount of skeletal advancement between two groups of patients with different factors” to “Compare horizontal skeletal relapse rates between two groups of patients with different factors”

- Revised P-value after the ANCOVA tests was used.

- Notations at the end of the table were revised.

4. (Page 19) In the discussion, part “Factors affecting skeletal relapse after cleft orthognathic surgery”, paragraph 3, we focused on the factor CO vs DO. Minor contexts were refined by adding the mean amount of skeletal advancement between CO vs DO and isolated maxillary vs bimaxillary surgery)

2. Furthermore, the rationale behind the chosen number of patients must be thoroughly described in the manuscript, rather than simply displaying the figures in the comments.

Thank you very much for the kind reminder. We have made revisions accordingly. The rationale behind the chosen number of patients has been added in the revised manuscript (Page 4-5).

3. From Section Reviewer comments to author, Topic 2.

Is the manuscript technically sound, and do the data support the conclusions?

Reviewer #1: Partly

Other than the revision made above, we refined the conclusion between CO group vs DO group to accurately reflect the results. (Page 25)

Reviewer #2: The study attempted to look for factors associated with skeletal relapse after maxillary advancement in cleft patients along with the correlation with soft tissue relapses. The study seems to be properly planned and executed and would serve as a valuable addition to the literature pertaining to cleft and craniofacial surgery.

Thank you kindly. We really appreciate you taking the time to share your comments with us.

---

## [Decision Letter · Decision Letter 2]

25 Oct 2023

Factors affecting the relapse of maxilla and soft tissues of nose, upper lip and velopharyngeal structures after maxillary advancement in cleft patients

PONE-D-23-08498R2

Dear Dr. Boonsiriseth,

We’re pleased to inform you that your manuscript has been judged scientifically suitable for publication and will be formally accepted for publication once it meets all outstanding technical requirements.

Kind regards,

Essam Al-Moraissi

Academic Editor

PLOS ONE

Additional Editor Comments (optional):

Reviewers' comments:

Reviewer's Responses to Questions

**Comments of Academic Editor**:

One or more reviewers has recommended that you cite specific previously published works. I recommend that you please review and evaluate the requested works to determine whether they are relevant and should be cited. It is not a requirement to cite these works.

**Comments to the Author**

1. If the authors have adequately addressed your comments raised in a previous round of review and you feel that this manuscript is now acceptable for publication, you may indicate that here to bypass the “Comments to the Author” section, enter your conflict of interest statement in the “Confidential to Editor” section, and submit your "Accept" recommendation.

Reviewer #3: (No Response)

Reviewer #4: All comments have been addressed

2. Is the manuscript technically sound, and do the data support the conclusions?

Reviewer #3: Yes

Reviewer #4: Yes

3. Has the statistical analysis been performed appropriately and rigorously? 

Reviewer #3: No

Reviewer #4: Yes

4. Have the authors made all data underlying the findings in their manuscript fully available?

Reviewer #3: Yes

Reviewer #4: Yes

5. Is the manuscript presented in an intelligible fashion and written in standard English?

Reviewer #3: Yes

Reviewer #4: Yes

6. Review Comments to the Author

Reviewer #3: The title of this manuscript is interesting; however, some flaws need to be addressed.

The abstract must have material/result/discussion/conclusion

The abstract/result section suffers from the lack of numerical values

The aim of the study must be written clearly at the end of the introduction.

There are too many tables that are not necessary, and some of them must be combined

The gender of patients must be clarified.

The type of malocclusion must be identified.

Were the samples vertical or horizontal growers?

The authors must provide a table to answer the two comments mentioned above, and at least the following cephalometric indexes must be mentioned.

1- Initial of treatment / 2- After surgery/ 3- After finishing the case/ 4- After retention period. This table is mandatory.

SNA, SNB, ANB, Wits appraisal, GoGn/SN

The "Discussion" section needs to be expanded. To expand the discussion, please cite the following articles.

Cephalometric changes in nasopharyngeal area after anterior maxillary segmental distraction versus Le Fort I osteotomy in patients with cleft lip and palate. Eur J Dent. 2018 Jul-Sep;12(3):393-397. doi: 10.4103/ejd.ejd_374_17.

The limitation of this study must be mentioned.

Reviewer #4: All questions have been adequately addressed and I would recommend this article for publicaiton. The relatively low subject numbers and the heterogeneity of the treatment methods are related to the nature of the deformtity. Hence optimal statistics to derive a conclusion is important. The authors elaborated this point.

7. PLOS authors have the option to publish the peer review history of their article (what does this mean?). If published, this will include your full peer review and any attached files.

Reviewer #3: **Yes: **Abdolreza Jamilian

Reviewer #4: No

---

## [Editor Report · Acceptance letter]

30 Oct 2023

PONE-D-23-08498R2 

Factors affecting the relapse of maxilla and soft tissues of nose, upper lip and velopharyngeal structures after maxillary advancement in cleft patients 

Dear Dr. Boonsiriseth:

I'm pleased to inform you that your manuscript has been deemed suitable for publication in PLOS ONE. Congratulations! Your manuscript is now with our production department. 

Kind regards, 

on behalf of

Dr. Essam Al-Moraissi 

Academic Editor

PLOS ONE